# CSA-LIC: Chroma Superpixel Aggregation for Machine-Oriented Learned Image Compression

## Abstract

Image compression for machines aims to remove redundancies in images while minimizing degradation in machine vision performance. However, existing methods use identical compression strategies for luma and chroma components, ignoring their perceptual differences in machine vision. To address this issue, a Chroma Superpixel Aggregation-based Learned Image Compression (CSA-LIC) method is proposed in this paper, which processes luma and chroma components differently according to their perceptual importance, and removes redundancies by exploiting intra-chroma and luma-chroma inter-component correlations. Specifically, a chroma adaptive sampling coding strategy is proposed, in which a superpixel-based chroma sampling module is designed to reduce chroma data volume by adaptively aggregating region-level semantic information based on chroma similarity, and a chroma generation module is built to enhance color integrity via luma compensation, thereby improving reconstructed chroma quality. To further eliminate cross-component redundancies, a cross-component feature transform module is designed to exploit luma-chroma inter-component correlations. Experimental results demonstrate that CSA-LIC outperforms state-of-the-art image compression methods in compression efficiency.

## 1 Introduction

With the rapid advancement of the Internet of Things (IoT) and deep learning technologies, Machine-to-Machine (M2M) communication accounts for a growing proportion of Internet traffic in applications such as autonomous driving (Li et al., 2024; Wu et al., 2025; Huang et al., 2025), video surveillance (Liu et al., 2025), and action recognition (Fan et al., 2022; Li et al., 2025a;b). In M2M systems, edge devices continuously collect and transmit large volumes of image data to cloud or edge servers for machine-based analysis, including object detection (Ren et al., 2017; Dang et al., 2023) and instance segmentation (He et al., 2017; Dai et al., 2025). This creates significant challenges for efficiently transmitting and storing massive image data, especially under limited bandwidth. However, most existing image compression methods are primarily optimized for signal fidelity (Zou et al., 2022; Liu et al., 2023; Jiang et al., 2025; Feng et al., 2025) or human visual perception (He et al., 2022; Pan et al., 2025), and thus fail to meet the demands of machine intelligence. Therefore, there is an urgent need to develop advanced Image Compression for Machines (ICM) tailored to machine intelligence.

To compress images for intelligent analysis, traditional ICM methods (Fischer et al., 2021; Kwak et al., 2023) utilize pre-trained task networks (e.g., Faster R-CNN (Ren et al., 2017) and Mask R-CNN (He et al., 2017)) to extract salient spatial regions and adaptively adjust quantization parameters. Subsequently, these methods apply standard codecs, such as High Efficiency Video Coding (HEVC) (Sullivan et al., 2012; Pan et al., 2020) and Versatile Video Coding (VVC) (Bross et al., 2021; Yuan et al., 2024), to compress the transmitted images. However, these traditional methods cannot optimize the coding framework end-to-end, which hinders further improvements in coding efficiency. To address this limitation, learned ICM methods have been developed, which can be broadly categorized into two types: attention mechanism-based methods (Le et al., 2021b; Fischer et al., 2023; Peng et al., 2024; Li et al., 2025c) and task loss-based methods (Le et al., 2021a; Yang et al., 2024; Zhang et al., 2024; Fischer et al., 2025). Attention mechanism-based methods focus

on preserving critical spatial regions for machine vision, while task loss-based methods jointly optimize the framework by integrating predictions or intermediate features into the rate-distortion loss. However, these methods apply identical compression strategies to luma and chroma components, neglecting their perceptual differences in machine vision, thereby limiting coding efficiency.

To solve this problem, a Chroma Superpixel Aggregation-based Learned Image Coding (CSA-LIC) method is proposed in this paper, which performs differentiated processing for luma and chroma components based on their perceptual differences in machine vision. Since chroma components generally exhibit higher redundancies than luma components in machine vision, CSA-LIC eliminates redundancies by exploiting intra-chroma and luma-chroma cross-component correlations. To improve chroma coding efficiency, a Chroma Adaptive Sampling Coding (CASC) strategy is developed, which comprises a Superpixel-based Chroma Sampling Module (SCSM) and a Chroma Generation Module (CGM). The SCSM is designed to reduce chroma data volume via adaptive region-level semantic aggregation, and the CGM is built to enhance reconstructed color integrity through luma compensation. To further eliminate cross-component redundancies between luma and chroma components, a Cross-component Feature Transform Module (CFTM) is designed to bridge spatial structure differences and extract cross-component correlations. Experimental results demonstrate that the proposed CSA-LIC achieves superior compression efficiency compared to state-of-the-art methods. The main contributions of this work are summarized as follows:

- We propose a CSA-LIC method that processes luma and chroma components differently according to their perceptual importance in machine vision, significantly improving coding efficiency.
- To effectively improve the coding efficiency of chroma components, a CASC strategy is developed, in which an SCSM is designed to reduce the chroma data volume via region-level semantic aggregation, and a CGM is built to enhance color fidelity through luma-guided compensation.
- To effectively remove cross-component redundancies, a CFTM is designed to align luma and chroma features and exploit their correlations.
- Experimental results demonstrate that CSA-LIC achieves superior compression efficiency compared to state-of-the-art image compression methods.

## 2 RELATED WORKS

### 2.1 TRADITIONAL ICM METHODS

To compress images for machine analysis, numerous traditional ICM methods (Shi & Chen, 2020; Fischer et al., 2021; Huang et al., 2021; Kwak et al., 2023; Kim et al., 2023) have been developed, demonstrating impressive compression performance. Shi & Chen (2020) transformed the bit allocation problem into a Markovian decision process, and introduced reinforcement learning to determine the quantization parameter of each Coding Tree Unit (CTU). Fischer et al. (2021) proposed a saliency-driven versatile video coding framework, in which a decision criterion based on salient regions is designed to identify salient CTUs for adaptive quantization parameter adjustment. Huang et al. (2021) introduced a visual analysis-motivated rate-distortion model, in which a CTU-level bit allocation strategy is developed according to the region of interest for machine, and a multi-scale feature distortion is designed to provide spatial context information. Kwak et al. (2023) developed a feature-guided machine-centric image compression method, which employs gradient maps between the original and reconstructed images to preserve task-relevant regions. Kim et al. (2023) introduced a machine-attention-based video compression method, which allocates higher bitrates to task-relevant regions through a maximum a posteriori-based bit allocation strategy. However, these methods fail to optimize the coding framework in an end-to-end manner, which limits compression efficiency.

### 2.2 LEARNED ICM METHODS

Learned ICM methods (Le et al., 2021a;b; Chen et al., 2023; Wang et al., 2023; Qi et al., 2023; Ahonen et al., 2023; Shindo et al., 2024a;b; Peng et al., 2024; Li et al., 2025c; Yin et al., 2025) optimize the compression framework in an end-to-end manner. Wang et al. (2023) developed a multi-task

collaborative optimization strategy that employs task loss to preserve critical semantic information. Qi et al. (2023) designed a saliency-guided bit allocation strategy, which allocates higher bitrates to key regions for improving semantic fidelity and task performance. Ahonen et al. (2023) introduced a region-of-interest image compression method, which employs a pre-trained task network to extract salient regions and refines feature representation via spatial attention. Shindo et al. (2024a) designed a region-guided mechanism to extract task-relevant features and incorporated a feature distortion loss for rate-distortion optimization. Shindo et al. (2024b) presented an edge structure-aware compression method, in which a segment-anything model is used to extract object edges, and an edge guidance mechanism is designed to preserve structure information. Peng et al. (2024) developed a saliency map-guided learned image compression method, in which a saliency map-guided mean square error loss is used to prioritize key spatial regions. Li et al. (2025c) implemented a spatial mask mechanism and a channel attention module to enhance task-relevant features across spatial and channel dimensions. Yin et al. (2025) developed a unified compression method for both human perception and machine vision, which introduces a contrastive language-image pre-training model to alleviate reliance on task networks. However, these methods fail to differentiate coding strategies between luma and chroma components according to their perceptual importance in machine vision, thereby limiting coding efficiency.

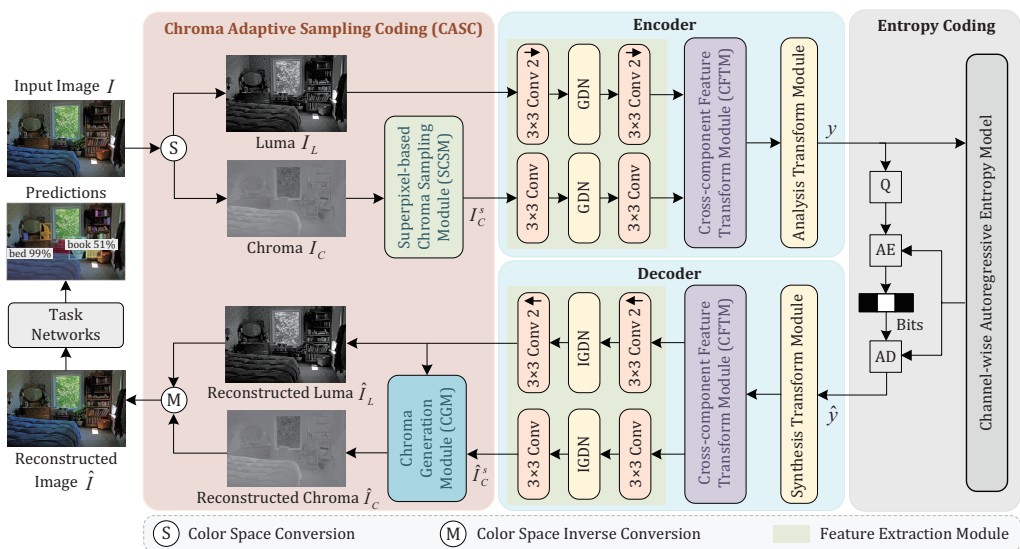

Figure 1: The overall architecture of the proposed CSA-LIC.

# 3 PROPOSED CSA-LIC

## 3.1 OVERALL ARCHITECTURE

To efficiently compress images for machine vision, we propose a CSA-LIC method, which employs differentiated processing strategies for luma and chroma components based on their perception differences in machine vision. The overall architecture of the proposed CSA-LIC is shown in Figure 1, which consists of four components: a CASC strategy, an encoder, an entropy coding, and a decoder. The CASC strategy is developed to effectively remove redundancies within chroma components, which contains an SCSM and a CGM. The SCSM is designed to reduce chroma data volume by aggregating region-level semantic information, and the CGM is built to enhance the quality of reconstructed chroma components by adaptively compensating chroma components with luma components. The encoder removes cross-component redundancies between luma and chroma components, in which a CFTM is designed to eliminate co-located spatial structure gaps and extract luma-chroma feature correlations. The entropy coding (Minnen & Singh, 2020) estimates latent feature distributions, while the decoder reconstructs the compressed luma and chroma components. The workflow of the proposed CSA-LIC is summarized as follows.

The input image $I$ is first transformed into luma components $I_L$ and chroma components $I_C$ via color space conversion (Cheng et al., 2001). To reduce chroma data volume, $I_C$ is processed by the SCSM to generate sampled chroma components $I_C^s$. The encoder jointly compresses $I_L$ and $I_C^s$ to learn compact latent features $y$. Specifically, a feature extraction module extracts luma features from $I_L$ and chroma features from $I_C^s$. These features are transformed by the CFTM to remove cross-component redundancies. The fused features are fed to an analysis transform module (Zou et al., 2022) to generate $y$. Next, $y$ are quantized to $\hat{y}$ and compressed using a channel-wise autoregressive entropy model (Minnen & Singh, 2020). The decoder reconstructs luma components $\hat{I}_L$ and chroma components $\hat{I}_C^s$ from $\hat{y}$. To improve the quality of reconstructed chroma components, $\hat{I}_C^s$ is enhanced by the CGM to obtain the final reconstructed chroma components $\hat{I}_C$. Finally, the reconstructed image $\hat{I}$ is obtained through color space inverse conversion of $\hat{I}_L$ and $\hat{I}_C$ for machine vision tasks.

## 3.2 CHROMA ADAPTIVE SAMPLING CODING (CASC)

Machine vision is particularly sensitive to structure properties present in luma components. The critical structure properties can effectively support machine intelligence even with constrained color information (Hou et al., 2020), indicating that chroma components contain higher redundancies than luma components. However, existing ICM methods (Chen et al., 2023; Peng et al., 2024; Yin et al., 2025; Fischer et al., 2025) generally adopt identical compression strategies for both luma and chroma components, neglecting their differential perceptual importance in machine vision, which consequently limits compression efficiency. To address this issue, we propose a CASC strategy to effectively remove chroma redundancies. The overall architecture of the proposed CASC is illustrated in Figure 1, which consists of an SCSM and a CGM. The SCSM aims to reduce chroma data volume via adaptive sampling, while the CGM enhances reconstruction quality through luma compensation.

### 3.2.1 SUPERPIXEL-BASED CHROMA SAMPLING MODULE (SCSM)

To reduce chroma data volume, an SCSM is proposed, which adaptively aggregates local regions based on chroma content similarity. The SCSM leverages a superpixel sampling network (Jampani et al., 2018) to merge chroma regions with consistent colors into unified superpixel representations, thereby effectively eliminating intra-region redundancies. Specifically, initial superpixel representations $\mathcal{S}^0$ are generated by averaging pixel values within regular grid regions. The module then computes associations between each chroma pixel and its neighboring superpixels to produce an association matrix. This matrix serves as pixel weights for iteratively updating the superpixel representations. After $n$ refinement iterations, the final superpixel representations $\mathcal{S}$ are obtained. The working process of the proposed SCSM is described as,

$$\begin{cases} \mathcal{S}^0 = \text{Avg}(I_C), \\ Q^t = \delta\left(\frac{I_C \otimes (\mathcal{S}^{t-1})^\top}{\sqrt{d}}\right), \\ \mathcal{S}^t = (Q^t)^\top \otimes I_C, \end{cases} \tag{1}$$

where $\text{Avg}(\cdot)$ represents the $4 \times 4$ average pooling; $\delta(\cdot)$ denotes the *softmax* function; $\otimes$ indicates the matrix multiplication; $Q^t$ represents the association matrix at the $t^{th}$ iteration; $\top$ denotes the matrix transposition; and $d$ indicates the channel dimension of chroma components.

To intuitively demonstrate the effectiveness of the proposed SCSM, Figure 2 compares original chroma components with their aggregated region-level representations. It can be seen that chroma components with similar characteristics are effectively clustered into coherent regions, such as the tie of the middle person. The visualization results strongly demonstrate that the proposed SCSM can effectively reduce chroma data volume, while significantly enhancing the representation capability of chroma components.

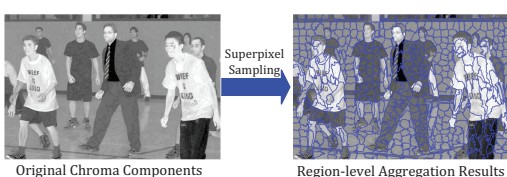

Original Chroma Components · Region-level Aggregation Results

Figure 2: An example of the original chroma components and its corresponding region-level aggregation results.

### 3.2.2 CHROMA GENERATION MODULE (CGM)

Superpixel-based chroma sampling leads to color information loss, degrading reconstructed image quality and task performance. To remove chroma redundancies while preserving the color integrity of chroma components, we propose a CGM, which adaptively compensates for chroma components using luma components, thereby improving the quality of reconstructed chroma components. The proposed CGM enhances both structure and content representations by extracting local structure and global semantic correlations between luma and chroma components. Figure 3 illustrates the architecture of the proposed CGM. To enhance

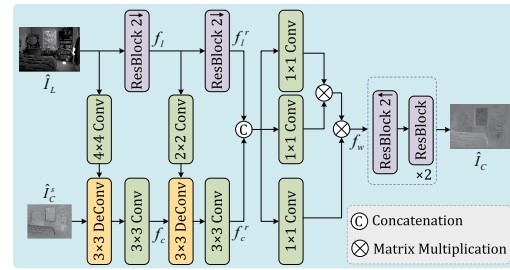

Figure 3: The architecture of the proposed CGM.

spatial structure consistency between reconstructed luma and chroma components, deformable convolution (Dai et al., 2017) is applied to adaptively mine local structure correlations. Specifically, luma features are first processed by a convolution layer to generate spatial offsets, which guide the deformable convolution layer to achieve precise spatial alignment. The proposed CGM effectively compensates for chroma spatial details through this adaptive receptive field adjustment guided by luma components. The aligned chroma features are then refined using a convolution layer. In parallel, luma features are processed through a down-sampling residual block (He et al., 2016) for feature refinement. Additionally, to achieve comprehensive content enhancement, we employ non-local attention mechanism (Vaswani et al., 2017) to model global semantic correlations between luma and chroma features. This attention mechanism adaptively extracts salient semantic information, thereby enhancing content integrity. Lastly, two up-sampling residual blocks refine the weighted chroma features to generate high-quality reconstructed chroma components $\hat{I}_C$. The working process of the proposed CGM is described as follows,

$$
\begin{cases}
f_c = \text{Conv}_3\Big(\text{DeConv}_3\big(\hat{I}_C^s, \text{Conv}_4(\hat{I}_L)\big)\Big), \\
f_l = \text{ResB}_{2\downarrow}(\hat{I}_L), \\
f_c^r = \text{Conv}_3\Big(\text{DeConv}_3\big(f_c, \text{Conv}_2(f_l)\big)\Big), \\
f_l^r = \text{ResB}_{2\downarrow}(f_l), \\
f_w = \text{NonAtten}\big(\text{Concat}[f_c^r, f_l^r]\big), \\
\hat{I}_C = \text{ResB}\Big(\text{ResB}_{2\uparrow}\big(\text{ResB}\big(\text{ResB}_{2\uparrow}(f_w)\big)\big)\Big),
\end{cases}
\tag{2}
$$

where $\text{Conv}_k(\cdot)$ and $\text{DeConv}_k(\cdot)$ denote the convolution and deformable convolution layers with a kernel size of $k \times k$, respectively; $\text{ResB}_{2\downarrow}(\cdot)$ and $\text{ResB}_{2\uparrow}(\cdot)$ represent the down-sampling and up-sampling residual blocks with a kernel size of $3 \times 3$ and a stride of 2, respectively; $\text{Concat}[\cdot]$ represents the channel-wise concatenation; $\text{NonAtten}(\cdot)$ denotes the non-local attention mechanism.

## 3.3 CROSS-COMPONENT FEATURE TRANSFORM MODULE (CFTM)

Due to the strong spatial and semantic correlations between luma and chroma components, their feature representations exhibit substantial cross-component redundancies. While feature fusion through luma-chroma correlation extraction can effectively reduce these redundancies, there exist spatial structure gaps between co-located luma and sampled chroma features. Conventional fusion methods (e.g., channel-wise concatenation) fail to establish effective inter-feature interactions, which severely limits redundancy elimination efficiency. To address this problem, a CFTM is designed, which bridges spatial structure gaps via dynamic cross-

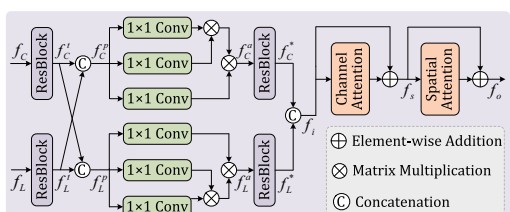

Figure 4: The architecture of the proposed CFTM.

component interaction and performs robust feature fusion through channel-spatial attention. The architecture of the proposed CFTM is illustrated in Figure 4.

To eliminate spatial structure gaps, a dynamic cross-component interaction is proposed to achieve co-located spatial structure alignment. Specifically, two residual blocks are first applied to enhance luma and chroma features through local receptive fields. Then, the enhanced luma and chroma features are concatenated in a channel-wise manner, and subsequently fed into three convolution layers to dynamically generate spatial mapping relationships through non-local interactions. Finally, two residual blocks are used to refine the aligned luma and chroma features. This progressive method effectively eliminates spatial structure gaps between luma and chroma components, while preserving their component-specific characteristics. Moreover, to effectively fuse the aligned luma and chroma features, a channel-spatial attention mechanism (Woo et al., 2018) is employed to extract luma-chroma correlations through channel and spatial attention, thereby eliminating cross-component redundancies. The working process of the proposed CFTM is formulated as follows,

$$
\begin{cases}
f_L^t = \mathrm{ResB}(f_L), f_C^t = \mathrm{ResB}(f_C), \\
f_L^p = \mathrm{Concat}[f_L^t, f_C^t], f_C^p = \mathrm{Concat}[f_C^t, f_L^t], \\
f_L^a = \delta\big(\mathrm{Conv}_1(f_L^p) \otimes \mathrm{Conv}_1(f_L^p)^\top\big) \otimes \mathrm{Conv}_1(f_L^p), \\
f_C^a = \delta\big(\mathrm{Conv}_1(f_C^p) \otimes \mathrm{Conv}_1(f_C^p)^\top\big) \otimes \mathrm{Conv}_1(f_C^p), \\
f_L^* = \mathrm{ResB}(f_L^a), f_C^* = \mathrm{ResB}(f_C^a), \\
f_i = \mathrm{Concat}[f_L^*, f_C^*], \\
f_s = \mathrm{ChanAtten}(f_i) + f_i, \\
f_o = \mathrm{SpatAtten}(f_s) + f_s,
\end{cases}
\tag{3}
$$

where $f_L$ and $f_C$ denote the luma and chroma features, respectively; $\mathrm{ResB}(\cdot)$ indicates a residual block with a kernel size of $3 \times 3$; $f_C^*$ and $f_L^*$ represent the aligned chroma and luma features, respectively; $\mathrm{ChanAtten}(\cdot)$ and $\mathrm{SpatAtten}(\cdot)$ mean the channel and spatial attention modules, respectively.

# 4 EXPERIMENTS

## 4.1 EXPERIMENTAL SETTINGS

### 4.1.1 DATASETS

The proposed CSA-LIC is trained on the COCO2017 training dataset (Lin et al., 2014). The original dataset contains 118,287 images, of which 117,465 with resolutions exceeding $256 \times 256$ pixels are retained for training. For each image, a $256 \times 256$ patch is randomly cropped as input to the CSA-LIC. We evaluate compression efficiency on the COCO2017 and OpenImagesV6 validation datasets (Kuznetsova et al., 2020), comparing the proposed method with other state-of-the-art image compression methods.

### 4.1.2 TRAINING SETTINGS

We use object detection and instance segmentation as machine vision tasks, employing Detectron2's Faster/Mask R-CNN X101-FPN (Wu et al., 2019) as the task network. CSA-LIC is optimized using rate-distortion loss, defined as,

$$
L = R + \lambda(D_i + D_f),
\tag{4}
$$

where $R$ represents the rate loss; $D_i$ denotes the pixel-level reconstruction loss between the reconstructed image $\hat{I}$ and original image $I$ using the Mean Squared Error (MSE) function; $D_f$ indicates the feature-level reconstruction loss from the task network's intermediate features using the MSE function; $\lambda \in \{0.001, 0.0025, 0.005, 0.01\}$ serves as the Lagrangian multiplier controlling the rate-distortion trade-off. We minimize this loss using the Adam optimizer (Kingma & Ba, 2015) with a batch size of 8 for 200 epochs. The learning rate follows a staged decay strategy: initially fixed at $1 \times 10^{-4}$ for the first 50 epochs, then halved every 30 epochs until reaching below $5 \times 10^{-6}$. The proposed CSA-LIC is implemented in PyTorch on an Ubuntu 20.04 platform with NVIDIA RTX 4090 GPUs.

## 4.2 RESULTS AND ANALYSIS

### 4.2.1 OBJECTIVE QUALITY ANALYSIS

To validate the effectiveness of the proposed CSA-LIC, we compare it with four state-of-the-art image compression methods: VVC-Intra (Zhang et al., 2021), LALIC (Feng et al., 2025), SMIC-Net (Peng et al., 2024), and UG-ICM (Yin et al., 2025). Compression efficiency is quantified using two metrics: Bjøntegaard Delta bitrate (BD-rate, $\eta_1$) for bitrate savings and Bjøntegaard Delta mean Average Precision (BD-mAP, $\eta_2$) at IoU threshold 0.5 for task performance improvements, both measured against the VVC-Intra anchor. As shown in Tables 1, CSA-LIC demonstrates superior performance across both object detection and instance segmentation tasks. On the COCO2017 dataset, it achieves average (BD-rate, BD-mAP) of (-16.068%, +3.781%), (-19.602%, +4.443%), and (-4.920%, +2.970%) against LALIC, SMIC-Net, and UG-ICM, respectively. Corresponding results on OpenImagesV6 are (-20.493%, +1.843%), (-30.548%, +2.682%), and (-3.460%, +1.257%). This advantage of the proposed CSA-LIC stems from two key innovations: (1) The SCSM reduces chroma data volume through region-level semantic aggregation, while the CGM enhances reconstruction quality via luma compensation. (2) The CFTM effectively removes cross-component redundancies to boost compression efficiency.

Table 1: Compression performance comparison in terms of BD-rate ($\eta_1$, %) and BD-mAP ($\eta_2$, %) on the COCO2017 and OpenImagesV6 datasets (anchor: VVC-Intra)

| Dataset | COCO2017 | | | OpenImagesV6 | | |
|---|---|---|---|---|---|---|
| Task | Detction | Segmentation | Average | Detction | Segmentation | Average |
| Method | $\eta_1/\eta_2$ | $\eta_1/\eta_2$ | $\eta_1/\eta_2$ | $\eta_1/\eta_2$ | $\eta_1/\eta_2$ | $\eta_1/\eta_2$ |
| LALIC | -32.370/4.417 | -28.942/3.623 | -30.656/4.020 | -24.147/1.402 | -28.698/1.821 | -26.423/1.612 |
| SMIC-Net | -29.121/3.732 | -25.122/2.983 | -27.122/3.358 | -14.476/0.555 | -18.259/0.990 | -16.368/0.773 |
| UG-ICM | -42.238/6.123 | -41.369/3.538 | -41.804/4.831 | -40.513/2.237 | -46.398/2.158 | -43.456/2.198 |
| **Proposed** | **-48.526/8.119** | **-44.922/7.482** | **-46.724/7.801** | **-43.345/3.221** | **-50.488/3.688** | **-46.916/3.455** |

For intuitive comparison of compression efficiency, Figure 5 presents the rate-accuracy curves of CSA-LIC versus other methods on both object detection and instance segmentation tasks. The results demonstrate that CSA-LIC consistently outperforms compared methods across these tasks. These findings indicate that the proposed method effectively minimizes performance degradation in machine vision tasks while achieving superior redundancy reduction. Furthermore, the consistent performance gains across diverse datasets validate the robustness and generalization capability of CSA-LIC.

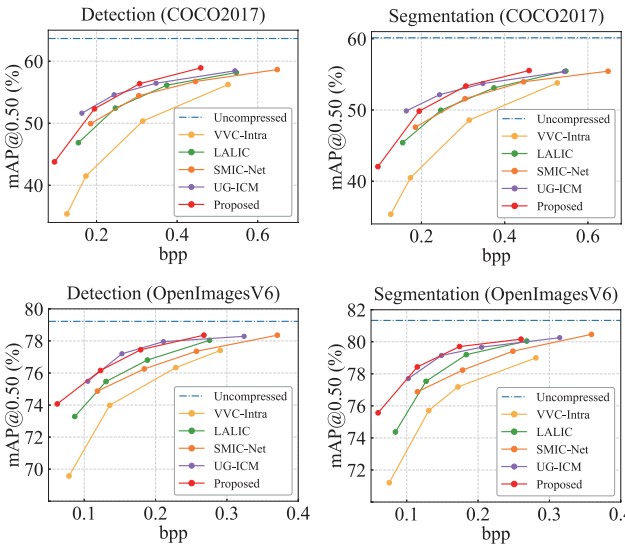

Figure 5: Rate-accuracy curves of each image compression method.

### 4.2.2 SUBJECTIVE QUALITY ANALYSIS

To visually verify the superiority of CSA-LIC, we select two representative samples from the COCO2017 dataset for qualitative analysis. Figure 6 compares the visual results of different methods on object detection and instance segmentation tasks. Notably, CSA-LIC achieves significantly higher accuracy while maintaining better bitrate efficiency (lower bpp) than other methods. These visual comparisons conclusively validate the dual-function capability of CSA-LIC, which achieves significant redundancy reduction while maintaining task performance without noticeable degradation.

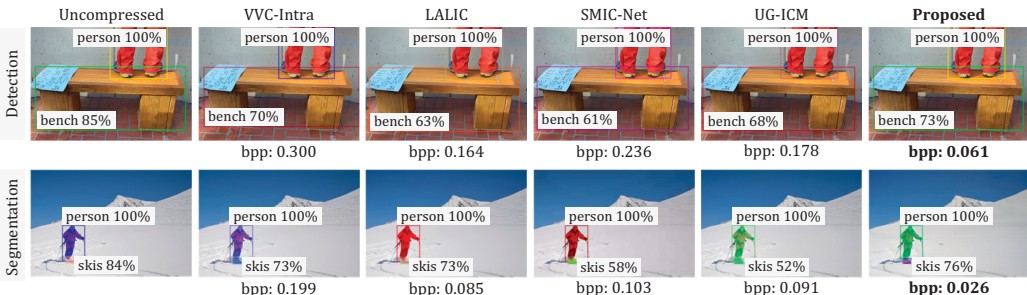

Figure 6: Qualitative comparison of object detection and instance segmentation performance on the COCO2017 dataset.

To further evaluate the reconstructed image quality for human perception, we select one representative image from the COCO2017 dataset for quality comparison. Figure 7 shows the reconstructed image quality of each image compression method. We can see that the proposed method achieves the lowest bpp of 0.298, significantly outperforming other methods in terms of compression efficiency. In terms of image quality, the proposed method achieves a PSNR of 23.143 dB, which is the second-highest among all the methods, closely following the LALIC method, which achieves the highest PSNR of 24.414 dB. However, the bpp of the proposed method is nearly half of LALIC's 0.530, illustrating its superior com-

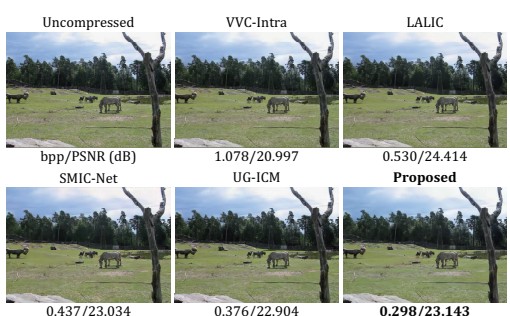

Figure 7: Reconstructed image quality for human perception.

pression performance without compromising image quality significantly. Overall, the proposed method offers an excellent balance between compression efficiency and reconstructed image quality, demonstrating a favorable trade-off compared to other image compression methods.

### 4.3 ABLATION ANALYSIS

The proposed CSA-LIC comprises two core components: CASC and CFTM. The CASC enhances chroma coding efficiency through two novel mechanisms: (1) an SCSM that reduces chroma data volume via adaptive region-level semantic aggregation, and (2) a CGM that improves reconstructed chroma quality through luma compensation. The CFTM eliminates cross-component redundancies by bridging spatial structure gaps and exploiting luma-chroma correlations. Ablation studies on the COCO2017 dataset (Table 2) validate each component's contribution: removing CASC degrades (BD-rate, BD-mAP) performance by (5.340%,

Table 2: Ablation study of CSA-LIC components: BD-rate ($\eta_1$, %) and BD-mAP ($\eta_2$, %), anchor: VVC-Intra

| SCSM | CFTM | CGM | Detection $\eta_1/\eta_2$ | Segmentation $\eta_1/\eta_2$ |
|------|------|-----|----------------|------------------|
| ✗ | ✗ | ✗ | -14.226/1.915 | -9.810/1.243 |
| ✗ | ✔ | ✗ | -43.186/7.338 | -40.165/5.873 |
| ✔ | ✗ | ✔ | -44.267/7.648 | -41.256/6.026 |
| ✔ | ✔ | ✔ | **-48.526/8.119** | **-44.922/7.482** |

0.781%) and (4.757%, 1.609%) for object detection and instance segmentation tasks, respectively, while replacing CFTM with channel-wise concatenation causes reductions of (4.259%, 0.471%) and (3.666%, 1.456%). These results conclusively validate the contributions of all components to CSA-LIC's superior coding efficiency.

To intuitively verify the effectiveness of the proposed CASC, we select one representative image from the COCO2017 dataset for quality comparison. The reconstructed images of the proposed CASC are presented in Figure 8. We can observe that the proposed CASC achieves bpp saving of 0.059, while yielding higher PSNR of 24.696 dB. These results strongly verify that its SCSM effectively removes chroma redundancies, and its CGM can improve the quality of reconstructed chroma components.

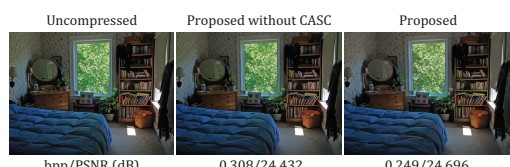

Figure 8: Reconstructed image quality of CASC.

To further validate the effectiveness of the proposed CFTM, Figure 9 compares feature visualization maps between channel-wise concatenation and CFTM. The results demonstrate that the proposed CFTM effectively eliminates cross-component redundancies while preserving salient spatial regions. This improvement stems from CFTM's dynamic cross-component interaction, which bridges spatial structure gaps between luma and chroma features. Additionally, it leverages robust feature fusion through channel-spatial attention mechanisms.

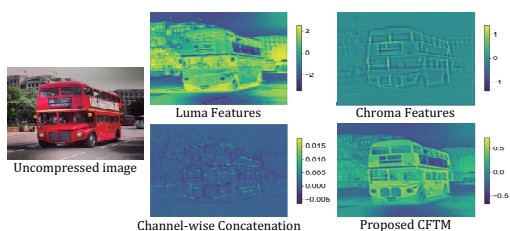

Figure 9: Feature visualization of the proposed CFTM.

### 4.4 COMPLEXITY ANALYSIS

We evaluate the computational complexity of CSA-LIC and other learned image compression methods by analyzing FLOPs, parameter counts, encoding time, and decoding time. As shown in Table 3, CSA-LIC achieves lower computational costs than compared methods: (1) Compared to LALIC and UG-ICM, it requires fewer FLOPs and parameters; (2) While SMIC-Net has reduced parameters and FLOPs, its compression performance is substantially worse; (3) Both encoding and decoding times

Table 3: Computational complexity comparison on the COCO dataset

| Method | FLOPs (G) | Parameters (M) | Encoding Time (s) | Decoding Time (s) |
|---|---|---|---|---|
| LALIC | 620.31 | 116.48 | 0.29 | 0.16 |
| SMIC-Net | 173.26 | 12.43 | 3.36 | 7.55 |
| UG-ICM | 360.34 | 117.78 | 0.22 | 0.28 |
| Proposed | 258.36 | 77.93 | 0.12 | 0.15 |

are significantly shorter than other methods. These results demonstrate that CSA-LIC provides superior compression efficiency with lower computational overhead across all metrics.

## 5 CONCLUSION

In this paper, we propose a CSA-LIC method that implements differentiated processing for both luma and chroma components to remove redundancies. The proposed CSA-LIC consists of two key components: CASC and CFTM. The CASC improves chroma redundancy elimination efficiency through an SCSM and a CGM. Specifically, the SCSM reduces chroma data volume through region-level semantic information aggregation, whereas the CGM enhances reconstructed chroma quality with luma compensation. The CFTM addresses spatial structure gaps between luma and chroma features by extracting luma-chroma correlations for eliminating cross-component redundancies. Extensive experiments demonstrate that CSA-LIC outperforms other state-of-the-art image compression methods.

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
