# OpenReview forum: "CSA-LIC: Chroma Superpixel Aggregation for Machine-Oriented Learned Image Compression"
_ICLR.cc/2026/Conference — ICLR 2026 Conference Withdrawn Submission_

### Official Review · Reviewer_rZ9M · 2025-10-29

**Soundness:** 2
**Presentation:** 2
**Contribution:** 2
**Rating:** 2
**Confidence:** 5

**Summary:**

This paper proposes a novel framework , CSA-LIC for Image Coding for Machines (ICM) designed to efficiently reduce redundancy for machine vision tasks by employing a separated processing pipeline for luma and chroma components. The core of this framework is the Chroma Adaptive Sampling Coding (CASC) strategy, which is integrated into the end-to-end nonlinear transform architecture. The CASC strategy includes a Superpixel-based Chroma Sampling Module (SCSM), which pre-processes the input by adaptively aggregating chroma regions to reduce data volume before the analysis transform, and a Chroma Generation Module (CGM), which post-processes the output by leveraging luma information to compensate and restore high-fidelity color details after the synthesis transform. To further enhance compression efficiency, a Cross-component Feature Transform Module (CFTM) operates within the codec, effectively eliminating redundancies between luma and chroma features. The author do experiments on the COCO2017 and OpenImagesV6 datasets for object detection and instance segmentation tasks to validate the proposed method, showing that it outperforms the baseline approaches discussed in the paper.

**Strengths:**

1. Drawing inspiration from traditional image coding principles, the authors identify the potential benefits of applying differentiated processing to luma and chroma components. This approach aims to eliminate information redundancy that is irrelevant to downstream machine vision tasks in luma and chroma spaces separately.
2. The paper provides empirical validation for its proposed modules. The authors have designed SCSM, CGM, and CFTM modules to implement their luma-chroma separation strategy and have conducted ablation studies that demonstrate performance gains over a baseline without these modules. This effectively validates the contribution of each component to the overall framework.

**Weaknesses:**

1. While the authors propose that a differentiated compression strategy for luma and chroma is beneficial, the paper lacks persuasive experimental analysis to substantiate this core claim. A foundational analysis demonstrating the varying impacts of luma and chroma degradation on different machine vision tasks performance would be crucial. Such an analysis would not only strengthen the motivation for this work but also provide a clearer justification for treating these components differently.
2.  The paper presents the framework more as a engineering integration rather than a principled design. The rationale behind key architectural choices is not adequately explained. For instance, why are SCSM and CGM designed as pre-processing and post-processing modules, respectively, outside the main codec, while CFTM is inserted within it? The paper can be more persuasive and readable if the author provide a deeper justification for this specific architecture over alternatives, such as modifying the nonlinear transform to handle luma and chroma features asymmetrically.
3. The design choice of the loss function appears suboptimal for the goal. The objective is to optimize performance on machine tasks, yet the loss function includes a pixel-level distortion term ($D_i$), which encourages fidelity to the original pixels, is often has a traed-off  with machine vision task performance. Including $D_i$ could inadvertently penalize solutions that are better for the task but have higher pixel-level fidelity, leading to significant performance drop in machine vision tasks.
4. The paper suffers from issues regarding its experimental setup. Firstly, the baseline codec used for ablation studies (\textit{i.e.}, the model without the proposed modules) is not clearly pointed out. Secondly, the comparison with other ICM methods may not be entirely fair. The proposed CSA-LIC model is trained in an end-to-end manner, while it is unclear if the competing methods received similarly optimized fine-tuning. A more comparable baseline would involve fine-tuning the base codec (without the proposed modules) on the target tasks with the same task-oriented loss function ($R + \lambda D_f$). Moreover, end-to-end fine-tuning may also implicitly learns the optimal luma and chroma importance allocation.
5. The literature review on ICM is not comprehensive. ICM, particularly the learning-based methods, is a rapidly evolving field with a vast amout of existing work. The authors' categorization of the field into "traditional" and "learning-based (attention- and task loss-based)" is an oversimplification. A more thorough investigation and discussion of recent, relevant papers are needed to properly contextualize the proposed method and differentiate its contributions.
6. The overall presentation and writing of the paper could be significantly improved. The manuscript currently reads as somewhat disorganized, and the logical flow between the different design components is not always clear. A thorough revision is recommended to enhance clarity, coherence, and the connection between the proposed modules. This is really important.

**Questions:**

As detailed in the weaknesses section, the manuscript has considerable room for improvement.

---

### Official Review · Reviewer_3iRg · 2025-10-29

**Soundness:** 4
**Presentation:** 4
**Contribution:** 3
**Rating:** 6
**Confidence:** 3

**Summary:**

As summarized in the abstract, a Chroma Superpixel Aggregation-based Learned Image Compression (CSA-LIC) method is proposed in this paper, which processes luma and chroma components differently according to their perceptual importance, and removes redundancies by exploiting intra-chroma and luma-chroma inter-component correlations. This method improves the existing methods which use identical compression strategies for luma and chroma components, ignoring their perceptual differences in machine vision. Experimental results demonstrate that CSA-LIC outperforms state-of-the-art image compression methods in different aspects.

I am not an expert in this area. Perhaps more results can be shown in supplementary materials to demonstrate the effectiveness of the proposed method.

**Strengths:**

As summarized in the abstract, a Chroma Superpixel Aggregation-based Learned Image Compression (CSA-LIC) method is proposed in this paper, which processes luma and chroma components differently according to their perceptual importance, and removes redundancies by exploiting intra-chroma and luma-chroma inter-component correlations. This method improves the existing methods which use identical compression strategies for luma and chroma components, ignoring their perceptual differences in machine vision. Experimental results demonstrate that CSA-LIC outperforms state-of-the-art image compression methods in different aspects.

**Weaknesses:**

I am not an expert in this area. Perhaps more results can be shown in supplementary materials to demonstrate the effectiveness of the proposed method.

**Questions:**

More results in supplementary materials may be helpful to demonstrate the effectiveness of the proposed method.

---

### Official Review · Reviewer_byzm · 2025-11-01

**Soundness:** 2
**Presentation:** 2
**Contribution:** 2
**Rating:** 2
**Confidence:** 4

**Summary:**

This paper proposes a Chroma Superpixel Aggregation-based Learned Image Compression (CSA-LIC) method for machine vision applications, which introduces a Superpixel-based Chroma Sampling Module (SCSM) that reduces chroma data volume through adaptive region-level semantic aggregation and a Chroma Generation Module (CGM) that enhances reconstructed chroma quality via luma compensation. Additionally, a Cross-component Feature Transform Module (CFTM) is designed to eliminate cross-component redundancies.

**Strengths:**

* The motivation of this paper is to explore the cross-component redundancy, which is similar to widely-used YUV420 color space in classical codec. It is reasonable and straightforward.
* Experimental results show that the proposed method can achieve good compression performance in terms of task accuracy.

**Weaknesses:**

There are several critical issues to be addressed.
* Some parts of the paper are not clearly described, making it difficult to understand the intended meaning. A thorough revision is recommended. For example, in the Related Work section, the authors categorize learned ICM into two types: attention mechanism-based methods and task loss-based methods. This classification is inappropriate, as these two categories are not mutually exclusive—many existing works utilize both approaches simultaneously.
* After applying SCSM, how exactly does the chroma data volume change? Does it involve a reduction in spatial dimensions or channel dimensions? The paper does not provide an explanation.
* In the proposed CSA-LIC framework, what are the specific structures of the analysis transform and synthesis transform? This information is missing in the paper.
* The experimental results are insufficient to convincingly demonstrate the effectiveness of the proposed method. It is recommended to include additional experimental results, refering to questions.

**Questions:**

* In Table 1, when compares the proposed method with other SOTA approaches, it is recommended to include more comparisons with machine-task-oriented compression methods. Currently, LALIC is designed for objective quality, not machine tasks, while UG-ICM is a hybrid method targeting both human perception and machine analytics.
* Besides, if authors would like to compare with sota PSNR-oriented methods like LALIC, it would be more fair to use the same loss function (MSE + task losses) and then illustrate the effectiveness of proposed CGM and CFTM.
* On the COCO2017 dataset, why does the proposed method exhibit performance degradation at low bitrates, performing worse than UG-ICM?
* In Figure 7, the authors claim that their method achieves similar PSNR performance at a lower bpp compared to LALIC, SMIC-Net and UG-ICM, therefore suggesting better compression performance. This claim is quite limited from one visualization of a specific image. For a fair comparison of objective quality, the authors should provide BD-rate (PSNR) results or RD curves in terms of PSNR.
* Figure 9 fails to demonstrate the effectiveness of CFTM, and its intended message is unclear. It is suggested to show the feature visualizations with CFTM and without CFTM to make it more clearly.
* Why is the encoding/decoding time of the proposed method much lower than that of SMIC-Net, despite having higher FLOPs and parameter counts?

---

### Note · Authors · 2025-12-25

I have read and agree with the venue's withdrawal policy on behalf of myself and my co-authors.